# IconDM: Text-Guided Icon Set Expansion Using Diffusion Models

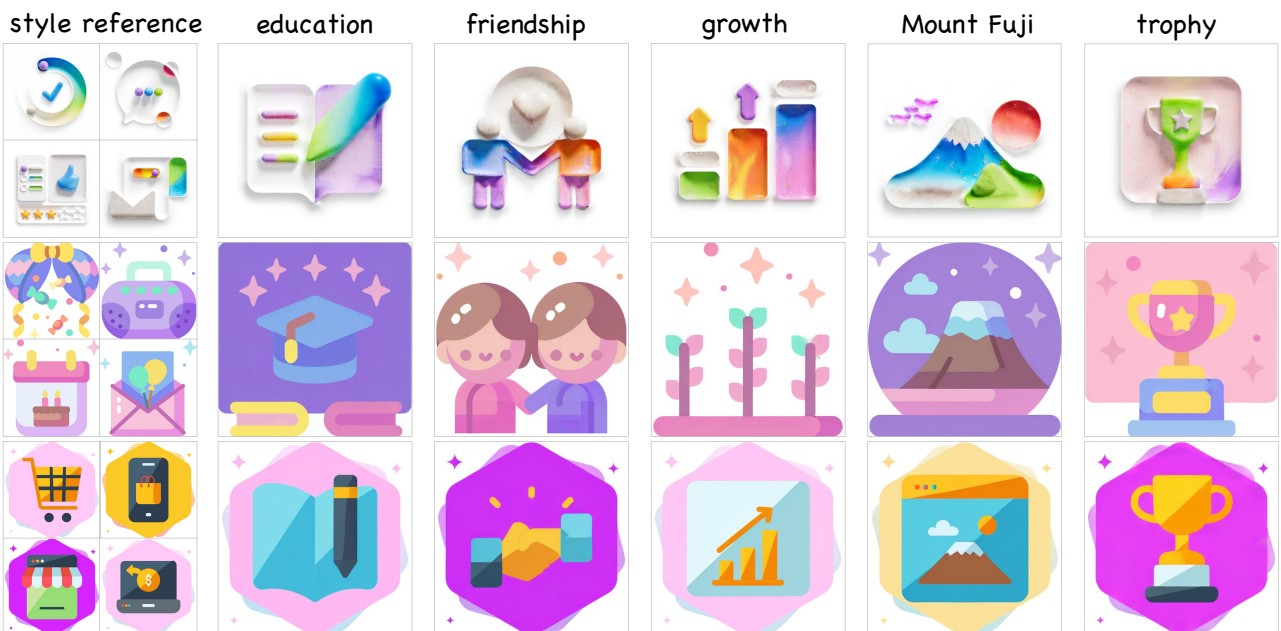

**Figure 1: Given a set of icons as style reference, IconDM can generate high-quality icons with consistent style features from textual descriptions, thereby expanding the initial icon set.**

## ABSTRACT

Icons are ubiquitous visual elements in graphic design. However, their creation is non-trivial and time-consuming. To this end, we draw inspiration from the booming text-to-image field and propose Text-Guided Icon Set Expansion, a task that allows users to create novel and style-preserving icons using textual descriptions and a few handmade icons as style reference. Despite its usefulness, this task poses two unique challenges. (i) Abstract Concept Visualization. Abstract concepts like *technology* and *health* are frequently encountered in icon creation, but their visualization requires a mental grounding process that connects them to physical and easy-to-draw concepts. (ii) Fine-grained Style Transfer. Unlike ordinary images, icons exhibit far richer fine-grained stylistic elements, including tones, line widths, shapes, shadow effects, etc, setting a higher demand on capturing and preserving them during generation.

To address the challenges, we propose IconDM, a method based on pre-trained text-to-image (T2I) diffusion models. It involves a one-shot domain adaptation process and an online style transfer

process. The domain adaptation aims to improve the pre-trained T2I model in understanding abstract concepts by finetuning on high-quality icon-text pairs. To achieve so, we construct IconBank, a large-scale dataset of 2.3 million icon-text pairs, where the texts are generated by the state-of-the-art vision-language model from icons. In style transfer, we introduce a Style Enhancement Module into the T2I model. It explicitly extracts the fine-grained style features from the given reference icons, and is jointly optimized with the T2I model during DreamBooth tuning. To assess IconDM, we present IconBench, a structured suite with 30 icon sets and 100 concepts (including 50 abstract concepts) for generation. Quantitative results, qualitative analysis, and extensive ablation studies demonstrate the effectiveness of IconDM.

## CCS CONCEPTS

• **Computing methodologies → Computer vision tasks**.

## KEYWORDS

icon generation, text-to-image, denoising diffusion models, style transfer

## 1 INTRODUCTION

We live in a world full of graphic design. Icons, the visual symbols encompassing a variety of styles and connotations, are widely used in graphic design of different contexts, such as branding, user interfaces, and way-finding systems. Compared to lengthy text, they

not only offer a more engaging visual representation, but also help users comprehend complex information in a more intuitive and effective manner. Additionally, icons have the ability to transcend language barriers, making them universally understood symbols.

The creation of icons, however, involves careful consideration of their visual appearance and expressiveness, making it a non-trivial and time-consuming task for human designers. Specifically, a well-designed icon should be expressive and easy to recognize, conveying its meaning at a glance. While this requirement can be met relatively easily for icons delivering concrete, physical concepts like *calendar*, it becomes challenging for those abstract and intricate concepts, such as *technology*, *system*, and *education*, as it requires a mental grounding process that connects these concepts to physical objects. For instance, *education* icons are typically visualized by an open book. Furthermore, to achieve visual harmony and enhance user experience, icons within a graphic design are expected to follow a particular design style and share common stylistic elements, including tones, line widths, shapes, shadow effects, etc (see the style reference in Figure 1). But unfortunately, it can be laborious to ensure style consistency, especially for large icon sets. Thus, such requirements bring a huge workload to human designers.

To ease the burdens of designers, we draw inspiration from the booming text-to-image field [4, 37, 41–43, 46, 55] and introduce a novel task, called *Text-Guided Icon Set Expansion*. Given a few handmade icons as style reference, the task aims to synthesize novel and stylized icons based on textual descriptions (see Figure 1). In this way, users can build a large set of style-coherent icons easily, without the tedious style-preserving creation process and the tough mental grounding process for abstract concepts.

While Text-Guided Icon Set Expansion closely relates to the booming fields of text-to-image and style transfer [9, 12, 37, 43, 45, 50], we find existing methods fall short in the task, primarily due to the following two reasons. (i) *Abstract Concept Visualization.* As mentioned earlier, abstract concepts are frequently encountered in icon creation. Although state-of-the-art text-to-image (T2I) diffusion models [37, 43] have achieved impressive results in generating high-quality images from text, they still struggle to ground abstract concepts to proper physical objects and render them in an icon-like style (see Figure 5). This is potentially caused by the lack of high-quality icon-text paired data in the training corpus of prevalent T2I models [47, 48]. (ii) *Fine-Grained Style Transfer.* Unlike ordinary image styles, icon styles contain extremely rich fine-grained stylistic elements, such as shapes, colors, shadows, as shown in Figure 1. As a result, transferring icon style is much more challenging. While previous approaches [9, 12, 45, 50] have successfully synthesized novel images that follow user-provided styles, directly applying them to icons leads to poor style preservation, especially for those fine-grained stylistic elements, as we will show in Section 4.

In this paper, we propose *IconDM*, a method based on pre-trained T2I diffusion models for Text-Guided Icon Set Expansion (see Figure 2). IconDM involves a one-shot domain adaptation process and an online style transfer process. In domain adaptation, IconDM aims to improve the base T2I model to understand abstract concepts and generate icon-like images. To achieve this, we construct *IconBank*, a large-scale dataset consisting of 2.3 million icon-text pairs crawled from the Internet, and finetune the pre-trained T2I model on it. Since icons from the Internet are only paired with a few concise

keywords, and directly using such data to finetune the T2I model may affect performance [1, 6], we utilize a state-of-the-art vision-language model (LLaVA [27]) to generate detailed descriptions for icons and finetune the model with a proper mix of concise keywords and detailed descriptions. In style transfer, IconDM takes a icon set with a few handmade, style-consistent icons as input and learns to synthesize novel while style-preserving icons. Here, we adopt the state-of-the-art personalization method DreamBooth [45] with Low-Rank adaptation (LoRA) [18]. While DreamBooth tuning is capable of grasping some coarse-grained stylistic elements from reference, it struggles to capture those fine-grained ones, partly because it does not have an explicit modeling of reference icons [60]. To address this limitation and better capture the fine-grained stylistic elements, we introduce a light-weight Style Enhancement Module (SEM) into the T2I model. It explicitly extracts style features from the reference icons, and is jointly optimized with the T2I model during DreamBooth tuning.

To evaluate IconDM, we introduce *IconBench*, a structured suite consisting of 30 diverse icon sets and 100 concepts for generation (50 of them are abstract concepts). We implement IconDM on the basis of Stable Diffusion XL [37]. Both the qualitative and quantitative results on IconBench show that IconDM outperforms existing text-guided style transfer methods in generating faithful, aesthetic and style-preserving icons. Extensive ablation studies also demonstrate the effectiveness of the domain adaptation process in improving abstract concept visualization and the Style Enhancement Module in boosting style preservation, especially for fine-grained stylistic elements. Furthermore, we also show that IconDM seamlessly works with ControlNet [59] to offer more controllability.

In summary, our contributions are as follows:

- We introduce Text-guided Icon Set Expansion, a novel task that allows users to expand icon sets using text prompts and a few reference icons, streamlining the design process.
- We present IconDM, a method based on the pre-trained T2I diffusion model. It leverages a domain adaptation process and a style transfer process with a light-weight Style Enhancement Module to tackle the challenging task.
- We construct IconBank, a large-scale dataset consisting of 2.3 million icon-text pairs from the Internet, and introduce Icon-Bench, a structured test suite including 30 icon sets and 100 concepts for method evaluation.

## 2 RELATED WORK

**Text-to-Image Generation.** The recent success of diffusion denoising probabilistic models (DDPMs) [14] has brought new life to the field of text-to-image generation. Different from their GAN counterparts [2, 8, 10, 33, 40], DDPMs use a denoising U-Net [44] to gradually remove noise and obtain clean images from Gaussian noise. Thanks to the effective model architecture and large-scale (image, text) datasets [47, 48], current T2I models [4, 35, 37, 41, 43, 46] can generate fairly high-quality images based on text prompts. In addition to text-to-image, DDPMs have also been applied to other computer vision tasks and achieved impressive results, such as image editing [11, 22, 57], inpainting [25, 29], video generation [13, 16, 30], 3D shape rendering [26, 45] and so on. Notably, although these T2I models are good at synthesizing open-domain

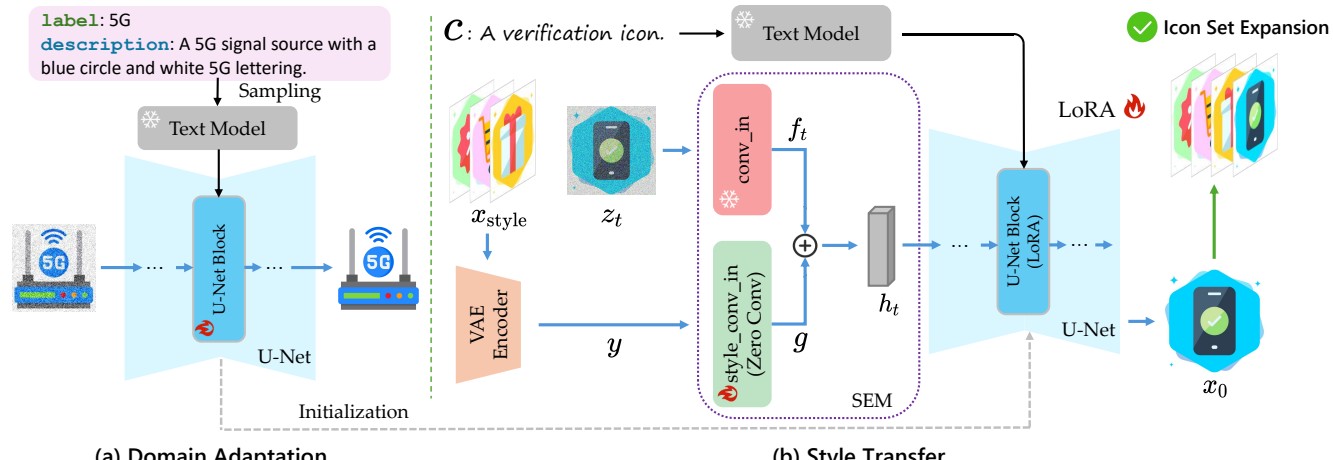

**Figure 2: An overview of the proposed IconDM. In domain adaptation, we leverage IconBank to finetune the full parameters of U-Net. The weights then serve as initialization for the style transfer process. To better achieve style preservation, we introduce a Style Enhancement Module (SEM) in style transfer, explicitly injecting the style features $g$ of reference icons $x_{style}$ into the model. After the two processes, IconDM can generate visually appealing, faithful and style-preserving icons $x_0$ based on the style reference icons $x_{style}$ and the text prompt $c$.**

images, they still have difficulty generating images in specific domains, such as icons. To overcome this limitation, in this work, we construct a large-scale icon-text dataset, named IconBank, and finetune the pre-trained T2I models on it for icon generation.

**Text-Guided Style Transfer.** Stylized image generation from text guidance has been widely studied, especially on T2I diffusion models [5, 9, 12, 34, 45, 50, 54, 60]. Existing methods can be grouped into 3 categories: (i) optimization-based [9, 24, 45, 50, 60]; (ii) optimization-free [5, 34, 38, 54]; and (iii) training-free [12, 53]. In particular, (i) optimization-based approaches finetune the model on a few reference images containing the target style. For example, Textual Inversion [9] optimizes the embedding of special tokens bound to the style while keeping all other parameters fixed. Dream-Booth [45] instead finetunes all the weights of U-Net [44], and it additionally introduces a prior preservation loss to avoid catastrophic forgetting of acquired knowledge. Some parameter-efficient finetuning techniques [17, 18] are adopted, seeking a trade-off between performance and computation resource. (ii) Optimization-free approaches [5, 34, 54], on the other hand, finetune a T2I model on large-scale stylized images such as WikiArt [20] for real-time stylized text-to-image generation. Due to the lack of pairwise (reference image, sample) data, they have to use the sample itself (or a cropped patch) as the reference image, potentially causing content leakage. To resolve this, the recent DEADiff [38] builds two paired datasets, and it proposes a content-style decouple mechanism and a non-reconstructive learning method for real-time style transfer. (iii) There are also training-free approaches that accomplish stylized image generation without any parameters tuning in offline or online. StyleAligned [12] introduces an attention-sharing operation with AdaIN [19] modulation during the diffusion process to maintain style consistency with the first image in the batch. InstantStyle [53] proposes to inject reference image features into style-specific blocks in U-Net so as to achieve effective style transfer without the need of

weight tuning. In our initial attempts, we found that optimization-based methods produce the highest quality icons, which drives us to design IconDM. To overcome existing methods' shortcomings in capturing fine-grained stylistic elements, we introduce a Style Enhancement Module (SEM) into the T2I model for explicit style feature modeling. In this way, the icons generated by IconDM have a more consistent style than the baselines.

**Icon Generation.** Icon Generation is an emerging topic in computer vision. Previous methods [32, 36, 58] typically utilize GANs to generate icons that satisfy specific class conditions. For example, Yang et al. [58] employ StyleGAN [21] to generate icons from 8 pre-defined categories, including weather, emotion, clothes and so on. They incorporate a self-attention mechanism and spectral normalization operation to enhance the quality and diversity of the generated icons. Chen et al. [7] propose IconGAN with dual discriminators to achieve icon generation conditioned on both app and theme labels. IconShop [56] adopts autoregressive Transformers [52] to synthesize scalable icons from texts, further improving practicality and flexibility. However, they suffer from the following shortcomings. First, previous approaches only consider semantic conditions and do not provide style control over the generated icons, thereby lacking the capabilities of producing icons in a desired style. Second, the work primarily focuses on concrete, physical concepts but does not delve into the generative capabilities of abstract concepts. In this paper, we present IconDM that for the first time allows users to generate new icons of either concrete or abstract concepts and control their styles to be consistent with reference icons.

## 3 ICONDM

In this section, we elaborate on IconDM for Text-Guided Icon Set Expansion. It is an optimization-based method built upon pre-trained T2I diffusion models. IconDM involves a domain adaptation and a

style transfer process (see Figure 2). Domain adaptation aims to improve the pre-trained T2I model in understanding abstract concepts and generating images in an icon-like style. To achieve so, we construct IconBank, a large-scale icon-text dataset, and perform extensive text revision with the state-of-the-art vision-language model. In style transfer, IconDM takes as input a few handmade icons for reference, and learns to synthesize novel while style-preserving icons using DreamBooth-LoRA [18, 45]. To better capture fine-grained stylistic elements in reference, we introduce a light-weight Style Enhancement Module into the T2I model.

In what follows, we first introduce the background of T2I diffusion models and then present the domain adaptation and style transfer process, respectively.

## 3.1 Preliminary

Diffusion denoising probabilistic models (DDPMs) are a class of generative models that have gained significant attention in recent years due to their ability to generate high-quality images. Given an input image $x_0$, their forward diffusion process gradually adds noise to the image to create a series of noisy images $x_1, x_2, ..., x_T$, where $T$ is the total number of steps in the diffusion process. The noisy image $x_t$ at each time step $t$ can be represented as a linear combination of the clean image $x_0$ and Gaussian noise $\epsilon \sim \mathcal{N}(0, \mathbf{I})$:

$$x_t = \sqrt{\bar{\alpha}_t}x_0 + \sqrt{1 - \bar{\alpha}_t}\epsilon. \qquad (1)$$

Here, $\bar{\alpha}_t = \prod_{i=1}^{t} \alpha_i$. And $\{\alpha_t\}_{t=1}^{T}$ denotes a pre-defined variance schedule, which guarantees that $\bar{\alpha}_T \approx 0$, so that the images $x_0$ will be diffused to a standard Gaussian noise $x_T \sim \mathcal{N}(0, \mathbf{I})$. In the reverse process, DDPMs are trained to progressively denoise the noisy image $x_t$ and obtain a cleaner image $x_{t-1}$ :

$$x_{t-1} = \frac{1}{\sqrt{\alpha_t}}(x_t - \frac{1 - \alpha_t}{\sqrt{1 - \bar{\alpha}_t}}\epsilon_\theta(x_t, t, c)) + \sigma_t\epsilon, \qquad (2)$$

where $\epsilon_\theta$ is a denoising neural network parameterized by $\theta$, $c$ is the condition for image generation and $\sigma_t$ reflects the noise intensity at time step $t$. In the implementation, $\epsilon_\theta$ typically adopts a U-Net [44] architecture. According to Equation 2, DDPMs can create new images from a random noise $x_T$ after $T$ iterations. With $x_t$ known as in Equation 1, the training objective of diffusion models is to minimize the $L2$ distance between the predicted noise $\epsilon_\theta(x_t, t, c)$ and its ground truth $\epsilon$. For stable training, the loss function is expressed as the following simplified formulation [14, 15]:

$$\mathcal{L}_{\text{ddpm}}^{\text{simple}} = \mathbb{E}_{t \sim [1,T], \epsilon \sim \mathcal{N}(0,\mathbf{I})} \left[ \|\epsilon_\theta(x_t, t, c) - \epsilon\|_2^2 \right]. \qquad (3)$$

Although DDPMs can synthesize high-fidelity images, their forward and reverse process both occur directly in pixel space, which can be computationally expansive during training and inference. To address this issue, latent diffusion models (LDMs) propose to diffuse and denoise in latent space. Specifically, they leverage a pre-trained variational auto-encoder (VAE) [23] to map an image $x$ to a low-dimensional latent vector $z$ using an encoder $\phi_{\text{enc}}$, i.e., $z = \phi_{\text{enc}}(x)$, and map it back using a decoder $\phi_{\text{dec}}$. Then, the forward process, reverse process and loss function can all be represented in a similar way to pixel space. For example, the loss function in latent space is:

$$\mathcal{L}_{\text{ldm}}^{\text{simple}} = \mathbb{E}_{t \sim [1,T], \epsilon \sim \mathcal{N}(0,\mathbf{I})} \left[ \|\epsilon_\theta(z_t, t, c) - \epsilon\|_2^2 \right]. \qquad (4)$$

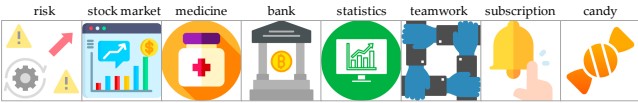

Figure 3: Some samples from IconBank.

| # icons | # sets | # icons per set | Avg. $L_{\text{label}}$ | Avg. $L_{\text{desc}}$ |
|---|---|---|---|---|
| 2,383,832 | 51,350 | 46.42 | 1.27 | 10.21 |

Table 1: The statistics of IconBank. Avg. $L_{\text{label}}$ represents the average word length of icon labels. Avg. $L_{\text{desc}}$ represents the average word length of icon descriptions.

## 3.2 Domain Adaptation

The purpose of domain adaptation is to finetune the T2I model for the icon domain, boosting its capability of understanding abstract concepts and generating images in an icon-like style. To accomplish this goal, we construct a large-scale icon-text dataset.

**IconBank.** As a dataset for domain adaptation, IconBank should meet the following two requirements. *(i) Diversity.* IconBank should contain icons in a wide variety of styles and also cover a wide range of concepts, from concrete, physical concepts to abstract, intricate concepts. *(ii) High-Quality Textual Description.* As highlighted in prior work [1, 6, 49], the quality of textual descriptions can largely impact the model performance.

To this end, we collect icons from Flaticon[1], one of the largest free online databases of manually designed icons. Icons in Flaticon are organized into sets, where the icons in a specific set share the same style. In addition, each icon in Flaticon is paired with a few concise textual labels, typically indicating its concept (see Figure 3). However, such concise labels or keywords may bring additional learning difficulty for T2I models, especially for abstract concepts. Hence, we follow the practice of previous work [6], and adopt the state-of-the-art vision-language model (LLaVA [27]) to generate detailed textual descriptions for icons. Empirically, we find that the instruction "*Please describe the icon in one sentence, the more detailed the better.*" can produce satisfactory results for LLaVA. An example of the generated icon description is shown in Figure 2a. Table 1 lists the statistics of the constructed IconBank. It contains 2.3 million icons from 51k sets and each icon is paired with a concise label and a detailed description. More examples and detailed procedures to construct the dataset are available in the supplementary materials. **Training.** We finetune the T2I diffusion model on IconBank with Equation 4. To support icon generation from either a concise label or a detailed description, we sample the text prompt $c$ for an icon from its label and description with equal probability.

## 3.3 Style Transfer

In style transfer, IconDM is given an initial set of icons $\mathcal{X}$ and learns to synthesize novel icons that share the same style with those in $\mathcal{X}$, including some fine-grained stylistic elements such as shapes, lines, 3D structure, shadows, etc. To meet this challenging requirement, the key insight of IconDM is to add explicit style

---

[1]https://www.flaticon.com/

guidance during model training and inference. This is achieved by a carefully designed Style Enhancement Module (SEM).

**Style Enhancement Module.** SEM uses a lightweight style convolutional layer to extract style features of reference icons and injects them into the denoising process of T2I model (see Figure 2b). Specifically, SEM takes as input $M$ icons $\{x^i_{\text{style}}\}^M_{i=1}$ as style reference from the initial icon set $\mathcal{X}$ and uses a VAE encoder to map them into the latent space:

$$y^i = \phi_{\text{enc}}(x^i_{\text{style}}), \tag{5}$$

where $y^i$ denotes the latent vector of $x^i_{\text{style}}$. Then, these resulting latent vectors $\{y^i\}^M_{i=1}$ are aggregated, fused with the sample's latent vector $z_t$ at time step $t$, and feeded into the U-Net for denoising:

$$f_t = \text{conv\_in}(z_t),$$
$$g = \frac{1}{M}\sum_{i=1}^{M}\text{style\_conv\_in}(y^i), \tag{6}$$
$$h_t = f_t + \lambda g.$$

Here, `conv_in` represents the original input convolutional layer of U-Net. `style_conv_in` is the aforementioned trainable style convolutional layer, which has the same parameter shapes as `conv_in`. $\lambda$ is a hyper-parameter that controls the style strength. When $\lambda = 0$, the model degenerates into a vanilla T2I model without additional style guidance. Finally, we obtain the fused feature $h_t$ and feed it into U-Net for denoising. Since $h_t$ contains style features, IconDM can explicitly take advantage of it to capture fine-grained style characteristics of reference icons during training and inference, and then generate new icons that are more consistent with the target style. For training stability, the added `style_conv_in` adopts a zero-initialization strategy [59], ensuring that its output is 0 at the beginning of training. This strategy helps prevent too drastic distribution shifts in the generation space.

**Training and Inference.** We employ the LoRA [18] variant of DreamBooth [45] to finetune the domain-adapted T2I model and the SEM module for style transfer. In each iteration, we sample a batch of training icons from $\mathcal{X}$ and pair each of them with $M$ reference icons. Note that the reference icons for each sample are guaranteed to exclude the training sample itself. In inference, given a text prompt, we sample $M$ reference icons from $\mathcal{X}$ and perform the denoising process to obtain a stylized icon.

## 4 EXPERIMENTS

### 4.1 Experimental Setting

**Implementation Details.** We adopt the state-of-the-art Stable Diffusion XL (SDXL) as our base text-to-image model. (i) In domain adaptation, we tune the full parameters of its U-Net [44] while keeping the parameters of the text encoder and VAE fixed, since they contribute little to domain adaptation. We conduct training on A100-80G GPUs with a batch size of 4, and employ AdamW [28] to optimize for about 500K iterations with a learning rate of 1e-5. Notably, domain adaptation is only performed once and does not require further adaptation to a specific icon set. (ii) In style transfer, the parameters of U-Net are first initialized from the resulting model of domain adaptation. Since we use the LoRA variant of DreamBooth, the model can be trained on a single 16G V100 GPU

with a batch size of 2 and a learning rate of 1e-4. We empirically find that 5,000 steps are sufficient to attain remarkable style transfer performance. As for hyper-parameters, the rank number of LoRA is set to 4, the number of style reference icons that are used in SEM is $M = 3$, and style strength is $\lambda = 1.0$. The icon resolution is set to $512 \times 512$. At inference, we adopt DDIM [51] sampler with 50 steps. For both domain adaptation and style transfer, we refer to the public Diffusers implementation[2]. In addition, we empirically find that the preservation loss in DreamBooth leads to unstable training, and hence, we disable it in our implementation.

**IconBench.** To evaluate the effectiveness of IconDM on Text-Guided Icon Set Expansion, we introduce a structured evaluation suite called IconBench. In particular, we randomly collect another 30 icon sets from Flaticon, which do not overlap with the icon sets in IconBank. With the help of ChatGPT, we collect 100 test concepts, including 50 concrete concepts and 50 abstract concepts. Please refer to the supplementary materials for the complete list of concepts. For each style and each concept, we generate 4 different icons, resulting in a total of $30 \times 100 \times 4 = 12000$ icons for evaluation.

**Evaluation Metrics.** We evaluate the generated results from two perspectives, namely text fidelity and style consistency. In terms of text fidelity, we measure it by the CLIP [39] cosine similarity between the generated icons and text prompts. For a fair comparison, all methods use the same text prompt *"A flat icon of the concept {}"* to compute CLIP text embeddings. To better understand the model's capability of visualizing abstract concepts, the CLIP score is further divided into CLIP (Concrete) and CLIP (Abstract), which are computed on concrete concepts and abstract concepts, respectively. As for style consistency, we leverage the DINO [3] score to evaluate it [12]. For each icon set, we calculate the average cosine similarity between DINO embeddings of the generated and reference icons. The calculations are again averaged over the 30 test icon sets. We do not use CLIP image embedding to measure style consistency since it focuses more on semantic-level features.

**Compared Baselines.** We compare IconDM against training-free, optimization-free and optimization-based approaches, respectively. Concretely, we choose StyleAligned [12] and InstantStyle [53] as the training-free baselines. We choose DEADiff [38] as our optimization-free baseline. Since optimization-based approaches are typically better at capturing fine-grained stylistic elements, we mainly compare against them, including Textual Inversion [9], DreamBooth [45], Custom Diffusion [24] and StyleDrop [50]. For StyleDrop, we use an unofficial implementation[3], since it is not open-source yet. For all other baselines except DEADiff (only has the SD-1.5 version), we adopt their SDXL version models for a fair comparison.

### 4.2 Main Results

**Qualitative Results.** Due to the space limitation, we only select the best qualitative results from each category of approaches for qualitative comparison (see Figure 4)[4]. The results indicate that (i) IconDM can visualize abstract concepts in a meaningful manner. For example, it uses a chip to represent *technology* and a bulb to

---

[2]https://github.com/huggingface/diffusers/
[3]https://github.com/aim-uofa/StyleDrop-PyTorch
[4]Please refer to the supplementary materials for a more comprehensive comparison

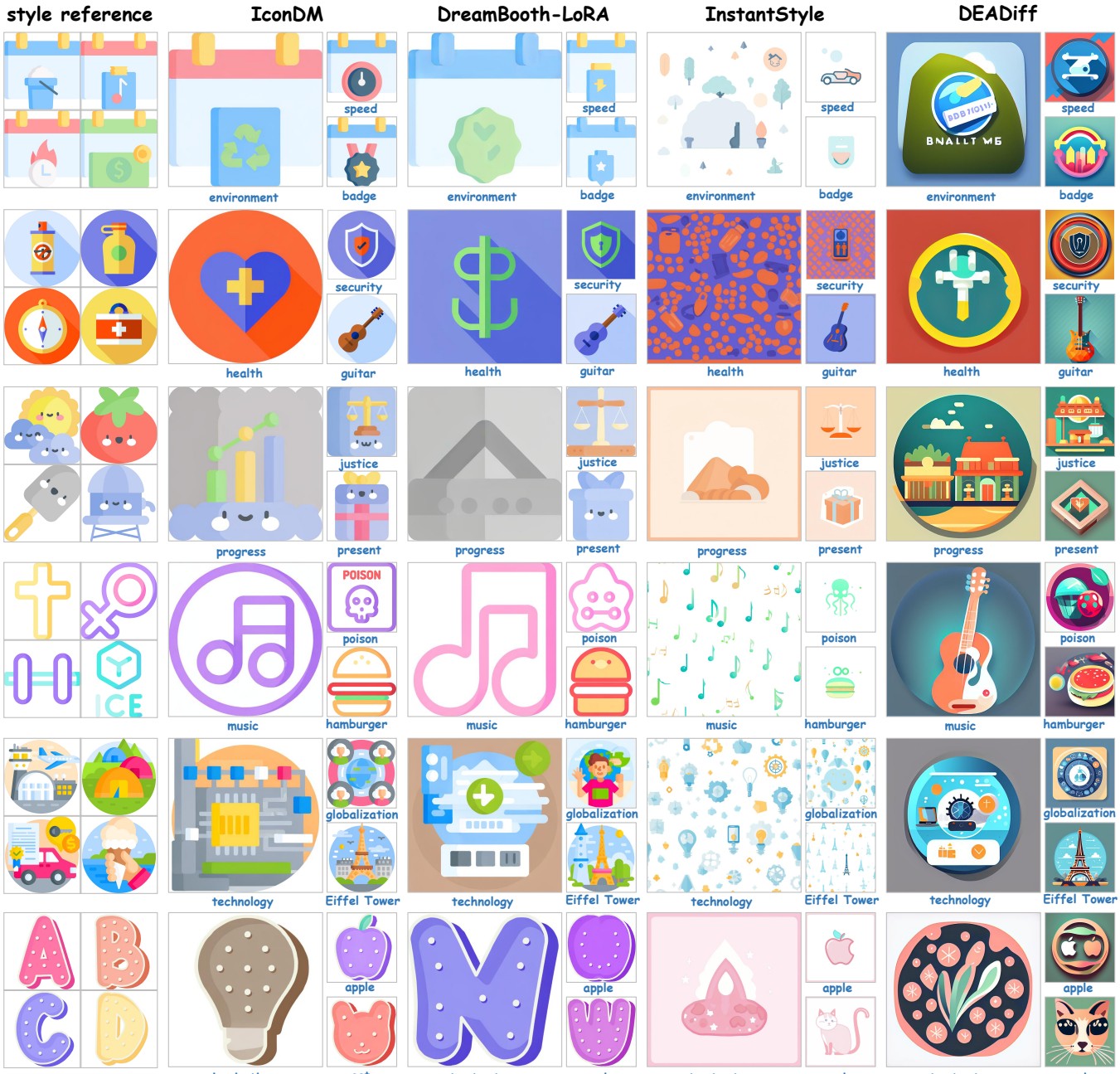

**Figure 4: Qualitative comparison between IconDM and other baselines. Please note that the gray borders around these icons are not generated by the models but are added later for a better visual experience.**

represent *inspiration*, allowing users to understand the underlying abstract concepts in an instant. (2) IconDM is proficient in fine-grained style transfer including color (the calendar color in the 1st row, color gradient in the 4th row), shape (round border in the 2nd row), line (line width in the 4th row), 3D effect (shadow in the 6th row) and even anthropomorphic effect (eyes in the 3rd row). In addition to these fine-grained stylistic elements, our approach also

ensures that the overall style of the generated icons is consistent with the reference ones, as evidenced by the 5th row. (3) Our approach can generate visually attractive icons. These three aspects of advantages allow IconDM to successfully address the aforementioned challenges and achieve Text-Guided Icon Set Expansion. However, the baseline methods all produce unsatisfactory results. Both the training-free method InstantStyle and optimization-free

| Methods | CLIP (Concrete) | CLIP (Abstract) | DINO |
|---|---|---|---|
| StyleAligned | 29.14 | 27.71 | 49.89 |
| InstantStyle | 31.94 | 28.01 | 45.65 |
| DEADiff | 31.29 | 28.45 | 42.33 |
| Textual Inversion | 31.01 | 25.91 | 30.22 |
| DreamBooth LoRA | 31.58 | 28.53 | 52.10 |
| Custom Diffusion | 31.98 | 28.59 | 51.59 |
| StyleDrop | 26.27 | 26.05 | 39.50 |
| Ours | **32.03** | **29.07** | **54.26** |

Table 2: Quantitative results of each method on Text-Guided Icon Set Expansion. These methods are divided into 4 parts, which are training-free, optimization-free, optimization-based approaches and our proposed IconDM from top to bottom of the table.

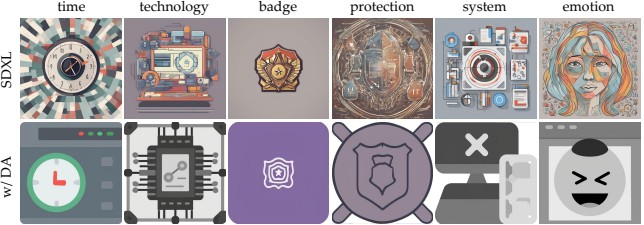

Figure 5: Qualitative comparison to demonstrate the effectiveness of domain adaptation (DA).

method DEADiff fail in fine-grained style transfer, and the icons they generate are of poor quality. Although DreamBooth-LoRA creates relatively higher-quality icons, it cannot solve the two challenges. First, it fails to generate some abstract concepts, such as *progress* in the 3rd row. Second, its capability of preserving fine-grained stylistic elements is limited. For instance, it fails to generate icons with a round border in the 2nd row.

**Quantitative Results.** Table 2 shows the quantitative results. From the results, we have the following observations. First, the CLIP (Abstract) score of all methods is lower than the CLIP (Concrete) score, which indicates that abstract concepts are more difficult to render, and their visualization is indeed a challenge in icon generation. Second, the results show that IconDM outperforms all baselines on all quantitative metrics, further proving that IconDM can generate high-quality icons that are faithful to the text prompts and are consistent with the styles of reference icons.

## 4.3 Ablation Studies

**Effect of Domain Adaptation.** Domain adaptation is a crucial process in IconDM. To investigate its effect, we conduct another experiment to directly perform style transfer on SDXL without domain adaptation (denoted as *w/o DA*). The quantitative metrics are shown in Table 3. From the results, IconDM achieves better CLIP (Concrete) and CLIP (Abstract) scores, which proves that the model gains better visualization capabilities for both concrete and

| Methods | CLIP (Concrete) | CLIP (Abstract) | DINO |
|---|---|---|---|
| Ours ($\lambda = 1.0, M = 3$) | 32.03 | 29.07 | 54.26 |
| w/o DA | 31.24 | 28.51 | 54.12 |
| $\lambda = 0.0$ (w/o SEM) | 31.79 | 28.91 | 53.57 |
| $\lambda = 0.5$ | 31.93 | 29.02 | 54.11 |
| $\lambda = 2.0$ | 31.96 | 29.09 | 54.68 |
| $M = 1$ | 32.00 | 29.01 | 54.24 |
| init. from conv_in | 31.31 | 28.71 | 54.06 |

Table 3: Ablation studies of IconDM.

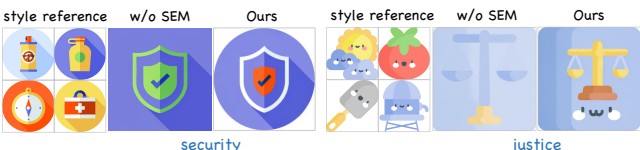

Figure 6: Qualitative comparison to demonstrate the effectiveness of Style Enhancement Module (SEM).

abstract concepts by training on the large-scale icon dataset. In addition, we qualitatively compare the icon generation performance of SDXL before and after domain adaptation (see Figure 5). The results indicate that (1) the icons generated by SDXL are sometimes not faithful to the textual descriptions. For example, it produces a high-definition face based on the text prompt "emotion", which is slightly off-topic, since the emotion represents the facial expression rather than the face itself. On the contrary, the icon generated by SDXL w/ DA skips the details of the face and highlights the facial expression, making it more aligned with the text prompt. (2) SDXL struggles to render abstract concepts in a meaningful manner. For instance, when the prompt "protection" is input into SDXL, it generates a poor quality icon with unrecognizable content. However, after domain adaptation, its variant SDXL w/ DA successfully visualizes the concept through a shield, allowing users to immediately understand the underlying concept. To sum up, both quantitative and qualitative results demonstrate the effectiveness of domain adaptation in icon generation. It not only provides the model with the capability to visualize abstract concepts, but also improves the fidelity between the generated icons and the input prompts.

**Effect of Style Enhancement Module.** Style Enhance Module (SEM) is introduced in the style transfer process to help the model better capture fine-grained style features. To study its effect, we disable SEM (by setting $\lambda = 0$, denoted as *w/o SEM*), degenerating the personalization method into ordinary DreamBooth-LoRA. This results in a drop in the DINO score, demonstrating that the style consistency between the generated icons and style reference icons is compromised without SEM. Additionally, we also show the qualitative results in Figure 6. Compared to IconDM, the w/o SEM variant lacks the capability to generate fine-grained styles. For example, in the 1st case of Figure 6, the variant does not capture the shape characteristics of the reference icons and fails to produce an icon with a round border. In the 2nd case, the variant ignores

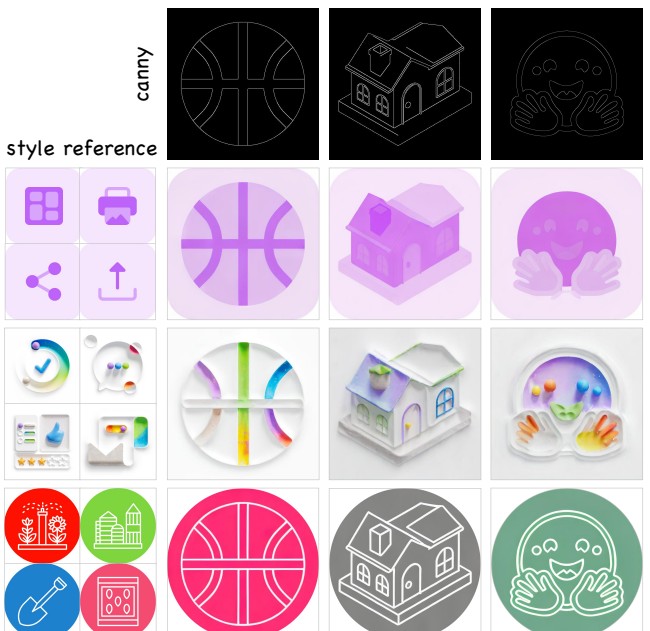

Figure 7: Feature visualization of the style convolutional layer and input convolutional layer.

Figure 8: With the support of ControlNet, IconDM can perform icon set expansion from canny images.

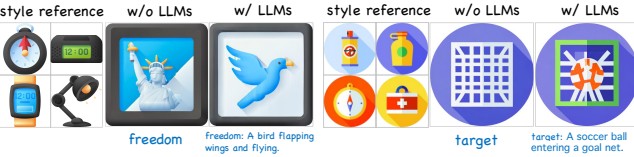

Figure 9: IconDM is integrated with LLMs to generate icons.

style_conv_in disrupt the feature space of the model, resulting in unstable training and affecting performance.

## 4.4 Feature Visualization

Thanks to the fact that the output of the convolutional layer does not destroy the spatial correspondence of the input image, we visualize the output features of the two convolutional layers involved in SEM, and provide a qualitative explanation for its effectiveness. To be more specific, we visualize the two output features at the last step in the reverse process (i.e. $t = 0$, near clean images), and the results are shown in Figure 7. We observe that the style convolution layer style_conv_in has significant output values on fine-grained stylistic elements, such as the 3D effect in case 1 and the shadow in case 2. In contrast, the output values of the input convolutional layer conv_in are average over the entire icons. This indicates that the added layer does pay more attention to the icon style, which qualitatively explains why our IconDM can achieve better fine-grained style transfer and preservation.

## 4.5 More Applications

**Combined with LLMs.** Given a concept label, we can prompt the pre-trained large language model (LLM) to generate a detailed description, and IconDM is capable of generating faithful and stylized icons accordingly (see Figure 9). In this way, IconDM can generate icons with more diverse content and richer details.

**Combined with ControlNet.** Our model can work seamlessly with ControlNet [59] to expand the icon set based on canny images. This is practical since designers sometimes not only need to ensure the semantic correctness of the generated icons, but also need to control the exact content of the icons. We show some generative icons in Figure 8. They are of remarkable quality while being stylistically consistent with the reference icons and faithful to the canny images. More qualitative results for other conditions can be found in the supplementary materials.

## 5 CONCLUSION

In this work, we propose a new task named Text-Guided Icon Set Expansion, aiming to help designers create new icons with controlled styles. To tackle this challenging task, we propose IconDM, decomposing it into two processes, i.e., domain adaptation and style transfer. The evaluation on IconBench demonstrates the effectiveness of our approach. Note that with the support of vectorization methods [31], IconDM can also generate icons in SVG format. Finally, we would like to point out that a limitation of our approach lies in its efficiency, i.e., the style transfer process is optimization-based and requires online learning for each icon set. In the future, we plan to seek an optimization-free method for Text-Guided Icon Set Expansion, thereby enhancing practicality.

the anthropomorphic eyes in the reference icons, leading to suboptimal style transfer performance. However, our IconDM can better preserve these fine-grained stylistic elements in generated icons. In summary, SEM is an effective module in icon style transfer. Finally, we investigate the effect of the hyper-parameter $\lambda$ in SEM. As $\lambda$ increases (from 0.0 to 2.0), the DINO score gradually gets improved (from 53.57 to 54.68). This shows that SEM does play a positive role in icon style transfer and preservation.

**Effect of Number of Reference Icons.** SEM in our approach can support any number of icons as style reference. To study the effect of the number, we use only 1 reference icon during training and inference of style transfer. From the quantitative results shown in Table 3, we find that the number of reference icons has little effect on the metrics. This also displays the excellent robustness of SEM.

**Effect of Initialization Strategy.** Since the style convolutional layer style_conv_in in SEM has the same parameter shape as the input convolutional layer conv_in of U-Net, we use the parameter of conv_in to initialize style_conv_in and investigate the effect of initialization strategy. The results are shown in Table 3. This non-zero initialization strategy harms all three quantitative metrics. We speculate this is because the non-zero initial hidden states of

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
