# OpenReview forum: "IconDM: Text-Guided Icon Set Expansion Using Diffusion Models"
_acmmm.org/ACMMM/2024/Conference — MM2024 Oral_

### Official Review · Reviewer_642T · 2024-05-15

**Rating:** 4
**Confidence:** 3

**Summary:**

This paper incorporates techniques from style transfer and T2I generation to construct a text-guided Icon set expansion following a set of style instructions. They introduce a large-scale dataset of 2.3 M icon-text pairs and an SDXL-based framework. The visualization results look impressive.

**Strengths:**

-The paper is well-written.

-A large-scale dataset is proposed.

-The visual results look good.

**Limitations:**

-It's unclear whether the online fine-tuning methods (like dreambooth) are implemented in the retrained unet on the Icon dataset or the original SD.

-The proposed method is not significantly better than the optimization-based approaches regarding CLIP metrics. Please explain it. Besides, taking the example (r2, c4) of Figure 1 as an example, I think this icon can not be matched well with 'growth' in CLIP space. Thus, a subjective assessment should be considered.

**Suitability:**

2

---

### Official Review · Reviewer_xhph · 2024-05-22

**Rating:** 3
**Confidence:** 4

**Summary:**

The paper presents IconDM, a method designed to expand icon sets using textual descriptions and a small set of handmade icons as style references. This approach leverages pre-trained text-to-image diffusion models to generate new icons that maintain the style characteristics of the reference icons. The task addresses two main challenges:

1. Abstract Concept Visualization: Converting abstract concepts (e.g., 'technology', 'health') into visual icons.
2. Fine-grained Style Transfer: Preserving detailed stylistic elements such as tones, line widths, shapes, and shadow effects in the generated icons.

**Strengths:**

1. Novelty: To the best of my knowledge, this is the first attempt to generate icons using a stable diffusion model. This innovative approach sets a new direction in the field of icon design.

2. Application: The task is highly relevant and useful in the industry. The authors demonstrate a strong understanding of industry requirements, which I greatly appreciate.

3. Writing: The paper is well-written, with clear motivations and explanations that make the content easy to follow.

**Limitations:**

I truly appreciate the innovation and effort put into this work. However, I have some concerns regarding the comprehensiveness of the evaluation.

1. Weakness: Lack of comparative evaluation with existing icon generation models

Justification: While this task is pioneering in the context of stable diffusion models, it is not the first attempt at icon generation in computer vision. It is crucial to compare the proposed method with existing icon generation models such as IconShop, IconGAN and etc, even though it is good to compare with relevant personalization stable diffusion models. Specifically based on your mentioned objectives, the evaluation should address:
(a) Style consistency: How well do other methods produce icons in the desired style compared to your approach?
(b) Abstract Concept Generation: How effective are other methods at generating abstract concepts relative to your method?

2. Weakness: Unclear definition of abstract concepts

Justification: In the Related Work, the paper mentioned other methods do not delve into the generative capabilities of abstract concepts. However, from my point of view, in IconGAN, they demonstrates the ability in generating abstract concept with an example like "setting". The paper also does not clearly define what constitutes a "good" abstract concept visualization or a benchmark/standard of an abstract concept visualization. For instance, an abstract concept like "technology" can be represented by various icons such as robots, chips or smartphones. A drone shop might use a drone icon to represent "technology", rather than a chip. Similarly, "environment" in your example could be visualized as green bushes as DreamBooth did, not only as a recycle icon. It is essential to establish the criteria for what makes an abstract concept visualization effective to avoid human-biased evaluations.


Thank you. All the best in your rebuttal and future research!

**Suitability:**

3

---

### Official Review · Reviewer_VoCo · 2024-05-24

**Rating:** 5
**Confidence:** 3

**Summary:**

The article presents a system inspired by text-to-image technology, Text-Guided Icon Set Expansion, which allows users to create novel and style-consistent icons using textual descriptions alongside a handful of hand-crafted icons as style references. Using a style transfer approach, the fine-tuning of Lora has yielded impressive results.

**Strengths:**

- The writing is clear and comprehensible. The authors present their narrative in a logical, fluid, concise, and accurate manner, which is excellent.
- Both the methods and evaluations are very solid.
- The results of the experiment look good.
- The questions are novel.

**Limitations:**

- A key issue here is who is the main user of this system and what user need is being met. There is a small contradiction as to whether the images generated are in a non-editable format. Obviously, if the images are not editable, professional designers cannot use them directly in their work. Designers need editable vector graphics (e.g. SVG) to create different versions for different applications, such as print form or screen form (mobile, tablet and web). As a result, this tool may not be an end-to-end solution for designers; instead, the content generated may only be used for inspiration or to showcase preliminary ideas. This tool seems more suitable for general users without design expertise. For example, a person opening a new online store with no design skills might need such an icon for their logo or other materials. Similarly, a software engineer designing icons for a new application would benefit from this tool. In the latter case, however, the software engineer would also need to modify the non-editable images to make them suitable for screen display. In summary, if this tool is to satisfy some of the user's needs, it should support the creation of vectorized or editable icons. However, given your current technical path (Lora), this may be a challenging problem to solve.
- Furthermore, if this tool were to become user-operable, supporting an interactive fitting method could satisfy a wider range of personalized needs. Users could select their desired model and refine the results generated by Lora with new prompts.
- Once the target user group and real-world application scenarios have been established, the author could consider conducting actual user testing of the system in subsequent research. This would allow for the collection of user feedback in order to iterate on the technology and its features.

**Suitability:**

3

---

### Meta-Review · Area_Chair_BAgu · 2024-07-10

**Recommendation:** Accept (Oral)
**Confidence:** 4

**Metareview:**

This paper received mixed ratings initially. After rebuttal, all the reviewers tend to accept the paper. SAC and AC agree with reviewers and recommend acceptance of the paper.